# Healthcare costs and outcomes associated with laboratory-confirmed Lyme disease in Ontario, Canada: A population-based cohort study

**Stephen Mac**[1,2]*, **Gerald Evans**[3,4], **Eleanor Pullenayegum**[5,6], **Samir N. Patel**[7,8], **Beate Sander**[1,2,3,7]

1 Institute of Health Policy, Management and Evaluation, University of Toronto, Toronto, Ontario, Canada, 2 Toronto Health Economics and Technology Assessment (THETA) Collaborative, University Health Network, Toronto, Ontario, Canada, 3 ICES, Toronto, Ontario, Canada, 4 Department of Medicine, Queen's University, Kingston, Ontario, Canada, 5 The Hospital for Sick Children, Toronto, Ontario, Canada, 6 Dalla Lana School of Public Health, University of Toronto, Toronto, Ontario, Canada, 7 Public Health Ontario, Toronto, Ontario Canada, 8 Department of Laboratory Medicine and Pathobiology, University of Toronto, Toronto, Ontario, Canada

* sm.mac@mail.utoronto.ca

**Data Availability Statement:** The dataset from this study is held securely in coded form at ICES. While legal data sharing agreements between ICES and data providers (e.g., healthcare organizations and

## Abstract

### Background

The objective of this study was to estimate the economic burden attributable to laboratory-confirmed Lyme disease (LD) in Ontario, Canada and assess health outcomes associated with LD.

### Method

We conducted a cohort study using laboratory-confirmed LD cases accrued between 2006 and 2018. The exposed cohort was matched 1:3 to the unexposed cohort using a combination of hard and propensity score matching. We used phase-of-care costing methods to calculate attributable costs for four phases of illness: pre-diagnosis, acute care, post-acute care, and continuing care in 2018 Canadian dollars. We used ICD-10-CA and OHIP billing codes to identify emergency department visits, physician billings and hospitalizations related to LD sequelae to assess health outcomes.

### Results

A total of 2,808 cases were identified with a mean age of 46.5 (20.7) years and 44% female. Within 30-days, 404 (14.3%) cases required an ED visit and 63 (2.4%) cases required hospitalization. The mean (95% CI) total costs for LD cases in pre-diagnosis, acute, and post-acute care phases were $209 ($181, 238), $1,084 ($956, $1,212), and $1,714 ($1,499, $1,927), respectively. The highest mean attributable 10-day cost was $275 ($231, $319) during acute care. At 1-year post-infection, LD increased the relative risk of nerve palsies by

government) prohibit ICES from making the dataset publicly available, access may be granted to those who meet pre-specified criteria for confidential access, available at www.ices.on.ca/DAS (email: das@ices.on.ca). The full dataset creation plan and underlying analytic code are available from the authors upon request, understanding that the computer programs may rely upon coding templates or macros that are unique to ICES and are therefore either inaccessible or may require modification.

**Funding:** This study was supported by ICES, which is funded by an annual grant from the Ontario Ministry of Health (MOH) and the Ministry of Long-Term Care (MLTC). This study also received funding from Canadian Institutes of Health Research (CIHR) project grant PJT149087 held by Beate Sander. Stephen Mac was supported by a CIHR Frederick Banting and Charles Best Canada Graduate Scholarship Doctoral Award GSD-159274. BS is supported by a Canada Research Chair in Economics of Infectious Diseases (CRC-950-232429). The funders had no role in study design, data collection and analysis, decision to publish, or preparation of the manuscript.

**Competing interests:** The authors have declared that no competing interests exist.

62 (20, 197), and polyneuropathy by 24 (3.0, 190). LD resulted in 16 Lyme meningitis events vs. 0 events in the unexposed.

## Conclusion

Individuals with laboratory-confirmed LD have increased healthcare resource use pre-diagnosis and up to six months post-diagnosis, and were more likely to seek healthcare services related to LD sequelae.

## Introduction

Lyme disease (LD) is the most commonly reported vector-borne disease in North America, caused by the spirochetal bacterium, *Borrelia burgdorferi*, which is transmitted from black-legged ticks [1]. In Canada, the annual number of LD cases has increased from 144 in 2009 to 2,636 in 2019 (18-fold increase) [2], and in Canada's most populous province (Ontario), the incidence rate has increased from 0.7 per 100,000 in 2010 to 7.9 per 100,000 persons in 2019 [3]. LD diagnoses in Ontario, Canada can be made clinically or through laboratory confirmation [3], and cases are reported to the integrated Public Health Information System (iPHIS).

The economic impact of LD in endemic parts of Europe and the United States are considerable compared to other vector-borne diseases. In a 2015 US study, LD resulted in an increase of $2,968 healthcare costs over 12-months compared to the controls [4]. Annual economic burden from a societal perspective in various countries was estimated to be between $0.14 to $786M US dollars [5]. In Canada, economic burden has not been well studied other than a recent study in 2019 by our colleagues estimating attributable LD costs to be $832 (2014 Canadian dollars) over one year [6]. Limited burden studies are partially due to the geospatial differences in risk of LD, and the relative novelty of the vector-borne disease in many provinces, resulting in the inability to recruit for prospective or retrospective cohort studies until recently. The long-term sequelae of LD are highly inconsistent across different populations, ages, and jurisdictions; and remain without consensus [7].

This study's objective was to estimate attributable healthcare costs in various LD phases-of-care, and health outcomes associated with laboratory-confirmed LD using a larger dataset, to further the understanding of economic and health burden to support the Federal Framework on Lyme Disease Act in Canada [8].

## Methods

### Study design and participants

We conducted an incidence-based, population-based matched cohort study using laboratory-confirmed LD cases in Ontario, Canada from the healthcare payer (Ministry of Health and Long-Term Care) perspective. Ontario has a publicly-funded universal healthcare system through the Ontario Health Insurance Plan (OHIP). We used person-level health administrative data housed at ICES. ICES is an independent, non-profit research institute whose legal status under Ontario's health information privacy law allows it to collect and analyze health care and demographic data, without consent, for health system evaluation and improvement [9]. This project has been approved by the Research Ethics Board at the Ontario Agency for Health Protection and Promotion (Public Health Ontario) and University Health Network. Data analyses were performed using SAS 9.4 (SAS Institute, Cary, NC).

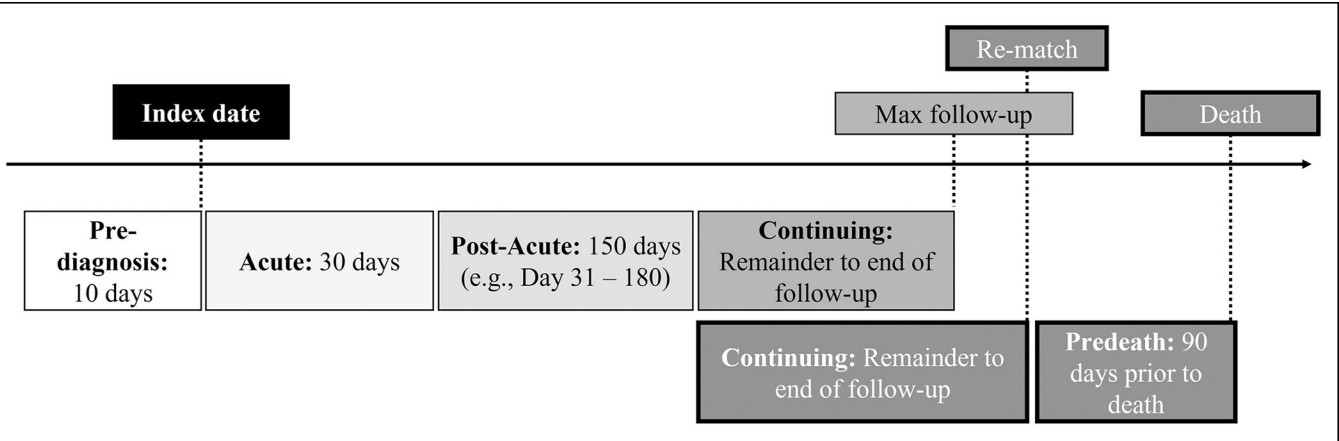

The timeline above illustrates the different phase-of-cares for Lyme disease by shade and border. If the individual dies, they have a predeath phase (bold border and darker shade) with the corresponding continuing phase calculated as shown (bold border and darker shade). An individual that does not die would have a continuing care phase length (light gray) until the maximum follow-up. For example, if someone dies within 90 days post-index date, then all of their time is assigned to the predeath phase. If someone dies on day 105, 90 days are assigned to predeath, with the remaining 15 days assigned to the acute phase.

**Fig 1. Phase-of-care for Lyme disease.**

We identified incident laboratory-confirmed LD cases between January 1, 2006, and December 31, 2018, using linked laboratory and reportable disease datasets, and LD case definitions (S1 File) [10]. Provincial reportable disease data is collected through iPHIS and serological testing results are collected through Public Health Ontario (PHO) [11], which is the only approved LD serological testing laboratory in Ontario. These datasets were linked using unique encoded identifiers and analyzed at ICES. Individuals were excluded if they were missing data related to age, sex, birth date, did not live in Ontario, ineligible for OHIP, or over 110 years of age at index date.

Index date was defined as the earlier of the positive PHO results date or iPHIS reportable disease date. Actual infection or disease onset date could not be identified and may have been before or on the index date depending on time of tick bite, timing of LD manifestations, and whether or not LD testing was considered [12]. We defined phase-of-care lengths by plotting mean total daily costs from 30 days prior to the index date to 365 days post-index date for all individuals and used joinpoint analysis to observe changes in healthcare costs suggestive of disease onset and progression [13, 14]. Through joinpoint regression analysis and advice from clinical experts (GE, SP), we defined five phases: pre-diagnosis (10-days prior to index), acute care (30-days post-index), post-acute care (31–180 days post-index), predeath (90-days prior to death date), and continuing care (time between end of post-acute care and predeath phase) (Fig 1).

## Propensity score and hard matching

We used a combination of propensity score matching and hard matching. We calculated propensity scores (i.e., likelihood for persons to be infected with LD) using a logistic regression on socioeconomic status, rurality, residing public health unit (PHU), and comorbidities. We used the Johns Hopkins ACG® System Aggregated Diagnosis Groups Version 10 to generate comorbidity profiles two years prior to index date [15]. The ACG is a person-focused diagnosis

based method of categorizing patients' illnesses using a combination of ICD-9 and ICD-10 codes and assigning to one of 32 diagnosis clusters. We used neighbourhood income quintiles as a proxy for socioeconomic status, rurality (Y/N) derived from Statistics Canada, and PHU as a proxy for risk of being an endemic area based on residence (S1 Table).

Individuals neither infected with LD nor had a PHO serologic test record were selected from the Registered Persons Database (RPDB) for matching. We excluded individuals with negative PHO test results to avoid potential confounding healthcare seeking behaviours. Each lab-confirmed LD case (exposed) was matched with up to three uninfected (unexposed) persons using a nearest neighbour, without replacement approach on sex, age ± 5 years, index year ± 1, and the logit of the propensity score using a caliper width of 0.2 standard deviations of the logit of the propensity score [16]. Weighted standardized differences were calculated to assess the balance using a threshold of 0.10 [17]. Since our accrued cohort spans over two decades from 2006 to 2018, there is a possibility of time-dependent changes to LD awareness and clinical management (e.g., cost of diagnosis, treatment), hence we included index year in matching.

To examine the effect of LD on costs before death, exposed individuals who died were matched with up to three unexposed persons (controls) who died during the observation period at 90 days prior to death. Exposed individuals were matched on sex, age ± 5 years, death date ± 90 days, and the logit of the propensity score.

## Outcomes and analysis

Costs were calculated following person-level costing methods established at ICES [18]. A summary of administrative datasets and brief descriptions are in S2 Table. We calculated and reported all mean and standardized 10-day costs in 2018 Canadian dollars. We estimated net attributable healthcare costs of laboratory-confirmed LD costs between exposed and unexposed individuals for each phase-of-care and cost categories using generalized estimating equation (GEE) analyses with the cost being the dependent variable, LD status as the independent variable, and matched sets treated as clusters. Each GEE analysis used a gamma distribution to inform variance parameterization and an identity link function [19, 20]. We used bootstrapping to resample matched sets with 1,000 replications to calculate 95% confidence intervals (95% CI).

We reported healthcare resource use and all-cause mortality for exposed individuals. ED visits and hospitalizations pre-index and post-index were determined using ICD-10 codes for LD and its related sequelae (S3 Table). Health outcomes associated with LD were estimated through healthcare seeking services (ED visits, physician billings, and hospitalizations) for conditions similar to known LD sequelae using the National Ambulatory Care Reporting System (NACRS), Discharge Abstract Database (DAD) and OHIP datasets (S2 Table), and a mix of ICD-10-CA and OHIP billing codes (S3 Table). Since there are no OHIP billing codes for LD; we used OHIP billing codes for conditions indicative of LD sequelae identified from a systematic review [7], and confirmed through clinical expert guidance (GE, SP). These sequelae included arthritis, carditis, cognitive (depression, delay in development), dermatologic, Lyme meningitis, polyneuropathy, and physical (headaches, nausea skin rashes) sequelae. We used a look back period of 1-year pre-index date to exclude sequelae that were also reported prior to confirmed LD infection (i.e., sequelae were reported pre-and post-index date were not considered attributable). We reported the proportion of the matched cohort developing sequelae and estimated relative risks (RR) using a GEE (poisson distribution, log link) to account for clustering.

All results are reported following the RECORD statement for observational studies (S4 Table) [21].

## Results

### LD cohort

A cohort of 2,808 individuals with laboratory-confirmed LD were included, mean (SD) age of 46.5 (20.7) years, and 44% female. Approximately 28% lived in rural areas, 54% were average to high socioeconomic status (i.e., neighbourhood income quintiles 3 to 5), and 61% of cases resided in four PHUs (Table 1). The age groups with most cumulative LD cases were 50 to 59, and 60 to 69 years. In age groups < 80 years, there were more male cases (S5 Table).

The annual incidence increased between 2006 and 2017 with males making up a majority in 2012 and onwards (Fig 2).

The mean (SD) and range of the follow-up time was 4.83 (3.06) years, and 0.15–14.29 years, respectively. Within 30-days of index date, 404 (14.3%) individuals required an ED visit and 63 (2.4%) were hospitalized with LD as the main responsible diagnosis. The mean (SD) length of stay per hospitalization was 5.5 (4.2) days. Within the exposed cohort, 40 (1.4%) individuals died of all-cause mortality at a mean (SD) time of 4.14 (3.01) years post-diagnosis (Table 2).

### Matched cohorts

After a combination of propensity score and hard matching at index date, 2,772 LD cases were matched to 8,217 individuals without LD (98.7%). Balance was assessed for all covariates and all weighted standardized differences were < 0.10, indicating good balance (Table 1). There were 34 cases who we were unable to match; these cases were more likely male (59%), resided in a single PHU (84%), resided in rural areas (75%), and in the highest neighbourhood income quintile (68%).

Among infected individuals who died, 38 (95%) were re-matched. The mean (SD) age at death for the re-matched exposed individuals was 72.2 (10.7) years. Nearly all weighted standardized differences were < 0.10, indicating a good balance (S6 Table).

### Healthcare costs

The mean (95% CI) costs for LD exposed individuals in the pre-diagnosis, acute, post-acute, and continuing care phases were $209 ($181, 238), $1,084 ($956, $1,212), $1,714 ($1,499, $1,927), and $11,013 ($9,854, $12,172), respectively while the mean (95% CI) costs for unexposed individuals were $96 ($81, $112), $260 ($226, $294), $1,339 ($1,221, $1,457) and $13,414 ($12,622, $14,208), respectively (S7 Table). Over a time period including the 10 days pre-diagnosis and 6-months post-diagnosis, the mean (95% CI) attributable cost of a laboratory-confirmed LD case was $1,312 ($1,166, $1,458). In 2019, where the LD incidence in Ontario was 7.9 per 100,000 persons (1,159 cases) [22], the estimated attributable 6-month cost to the healthcare system would have been approximately $1,520,608.

Mean (95% CI) attributable 10-day costs was highest in the acute care phase at $275 ($231, $319), with hospitalization being the largest component at $160 ($127, $192). Mean (95% CI) attributable 10-day costs for the pre-diagnosis phase was $113 ($81, $144) with hospitalizations and ED visit costs contributing most of the costs at $46 ($26, $67), and $39 ($33, $45), respectively. Fig 3 illustrates the total mean 10-day costs for the exposed and unexposed cohorts in each of the four phases of care, broken down by cost types (e.g., hospitalization costs, physician costs). Laboratory and drug costs were also highest in the pre-diagnosis and acute care phases of LD, but nearly negligible in the later LD phases of care (Table 3).

Mean (95% CI) attributable 10-day costs for the post-acute care, and continuing care phase was $20 ($2, $37), and -$22 ($-33, -$10), respectively suggesting that LD generally does not result in additional long-term healthcare costs. In the predeath care phase, the mean (95% CI) attributable 10-day cost of all-cause mortality post-LD infection was $391 (-$1,152, $1,934).

**Table 1. Baseline characteristics of matched cohort at index date.**

|  | Before matching | | After matching | | |
| --- | --- | --- | --- | --- | --- |
| Variables | Exposed individuals (n = 2,808) | Unexposed individuals (n = 369,250) | Exposed individuals (n = 2,772) | Unexposed individuals (n = 8,217) | Weighted Standardized Differences |
| **Age** | | | | | |
| Mean ± SD | 46.50 ± 20.65 | 40.35 ± 22.59 | 46.51 ± 20.58 | 46.37 ± 20.57 | 0 |
| Median (IQR) | 51 (31–62) | 40 (22–57) | 51 (31–62) | 51 (31–62) | 0 |
| **Sex** | | | | | |
| Female | 1,245 (44.3%) | 186,323 (50.5%) | 1,231 (44.4%) | 3,657 (44.5%) | 0 |
| Male | 1,563 (55.7%) | 182,927 (49.5%) | 1,541 (55.6%) | 4,560 (55.5%) | 0 |
| **Neighbourhood Income Quintile** | | | | | |
| 1 (lowest) | 325 (11.6%) | 74,020 (20.1%) | 325 (11.7%) | 985 (11.9%) | 0.00 |
| 2 | 466 (16.6%) | 72,803 (19.8%) | 464 (16.7%) | 1,298 (15.7%) | 0.02 |
| 3 | 606 (21.6%) | 73,093 (19.9%) | 600 (21.6%) | 1,829 (22.1%) | 0.01 |
| 4 | 589 (21.0%) | 74,203 (20.2%) | 586 (21.1%) | 1,769 (21.3%) | 0.01 |
| 5 (highest) | 819 (29.2%) | 73,912 (20.1%) | 797 (28.8%) | 2,410 (29.1%) | 0.02 |
| Missing | 0 (0.0%) | 93 (0.0%) | - | - | - |
| **Rural** | | | | | |
| No | 2,014 (71.8%) | 330,525 (89.7%) | 2,005 (72.3%) | 6,039 (72.8%) | 0.00 |
| Yes | 792 (28.2%) | 37,992 (10.3%) | 767 (27.7%) | 2,252 (27.2%) | 0.00 |
| **Public Health Unit** | | | | | |
| Algoma | 6 (0.2%) | 3,087 (0.8%) | 6 (0.2%) | 12 (0.1%) | 0.00 |
| Brant | 12 (0.4%) | 4,034 (1.1%) | 12 (0.4%) | 37 (0.4%) | 0.00 |
| Durham | 94 (3.3%) | 17,318 (4.7%) | 92 (3.3%) | 248 (3.0%) | 0.00 |
| Elgin-St Thomas | 7 (0.2%) | 714 (0.2%) | 7 (0.3%) | 23 (0.3%) | 0.00 |
| Bruce-Grey-Owen Sound | 11 (0.4%) | 4,248 (1.2%) | 11 (0.4%) | 39 (0.5%) | 0.02 |
| Haldimand-Norfolk | 36 (1.3%) | 2,914 (0.8%) | 36 (1.3%) | 125 (1.5%) | 0.00 |
| Haliburton-Kawartha-Pine Ridge | 47 (1.7%) | 4,786 (1.3%) | 47 (1.7%) | 157 (1.9%) | 0.00 |
| Halton | 59 (2.1%) | 14,605 (4.0%) | 59 (2.1%) | 188 (2.3%) | 0.01 |
| Hamilton-Wentworth | 41 (1.5%) | 14,759 (4.0%) | 41 (1.5%) | 116 (1.4%) | 0.02 |
| Hastings and Prince Edward | 151 (5.4%) | 4,365 (1.2%) | 151 (5.4%) | 485 (5.8%) | 0.01 |
| Huron | <6 (<0.2%) | 1,578 (0.4%) | <6 (<0.2%) | 6 (0.1%) | 0.00 |
| Kent-Chatham | 13 (0.5%) | 2,718 (0.7%) | 12 (0.4%) | 35 (0.4%) | 0.00 |
| Kingston-Frontenac-Lennox and Addington PHU | 445 (15.9%) | 5,043 (1.4%) | 441 (15.9%) | 1,345 (16.2%) | 0.00 |
| Lambton | <6 (<0.2%) | 3,511 (1.0%) | <6 (<0.2%) | 13 (0.2%) | 0.01 |
| Leeds-Grenville-Lanark | 505 (18.0%) | 4,339 (1.2%) | 478 (17.2%) | 1,403 (16.9%) | 0.00 |
| Middlesex-London | 34 (1.2%) | 12,333 (3.3%) | 34 (1.2%) | 112 (1.4%) | 0.01 |
| Niagara | 58 (2.1%) | 12,062 (3.3%) | 58 (2.1%) | 176 (2.1%) | 0.02 |
| North Bay | <6 (<0.2%) | 3,576 (1.0%) | <6 (<0.2%) | 15 (0.2%) | 0.01 |
| Northwestern | 23 (0.8%) | 2,293 (0.6%) | 23 (0.8%) | 71 (0.9%) | 0.02 |
| Ottawa Carleton | 461 (16.4%) | 25,762 (7.0%) | 461 (16.6%) | 1,285 (15.5%) | 0.01 |
| Oxford | 10 (0.4%) | 799 (0.2%) | 10 (0.4%) | 29 (0.3%) | 0.01 |
| Peel | 60 (2.1%) | 38,554 (10.5%) | 60 (2.2%) | 201 (2.4%) | 0.01 |
| Perth | 8 (0.3%) | 2,081 (0.6%) | 8 (0.3%) | 24 (0.3%) | 0.02 |
| Peterborough | 26 (0.9%) | 3,825 (1.0%) | 26 (0.9%) | 89 (1.1%) | 0.01 |
| Porcupine | <6 (<0.2%) | 2,325 (0.6%) | <6 (<0.2%) | 14 (0.2%) | 0.02 |
| Renfrew | 19 (0.7%) | 2,727 (0.7%) | 19 (0.7%) | 57 (0.7%) | 0.00 |
| Eastern Ontario | 130 (4.6%) | 5,544 (1.5%) | 130 (4.7%) | 361 (4.4%) | 0.02 |

*(Continued)*

**Table 1.** (Continued)

| Variables | Before matching | | After matching | | |
| --- | --- | --- | --- | --- | --- |
| | Exposed individuals (n = 2,808) | Unexposed individuals (n = 369,250) | Exposed individuals (n = 2,772) | Unexposed individuals (n = 8,217) | Weighted Standardized Differences |
| Simcoe | 56 (2.0%) | 14,604 (4.0%) | 56 (2.0%) | 179 (2.2%) | 0.00 |
| Sudbury | <6 (<0.2%) | 5,205 (1.4%) | <6 (<0.2%) | 21 (0.3%) | 0.03 |
| Thunder Bay | 6 (0.2%) | 4,256 (1.2%) | 6 (0.2%) | 16 (0.2%) | 0.00 |
| Timiskaming | <6 (<0.2%) | 952 (0.3%) | <6 (<0.2%) | 6 (0.1%) | 0.00 |
| Waterloo | 50 (1.8%) | 14,297 (3.9%) | 50 (1.8%) | 132 (1.6%) | 0.00 |
| Wellington-Dufferin-Guelph | 27 (1.0%) | 7,624 (2.1%) | 27 (1.0%) | 90 (1.1%) | 0.01 |
| Windsor-Essex | 25 (0.9%) | 11,198 (3.0%) | 25 (0.9%) | 88 (1.1%) | 0.01 |
| York | 78 (2.8%) | 30,152 (8.2%) | 78 (2.8%) | 216 (2.6%) | 0.00 |
| City of Toronto | 288 (10.3%) | 76,544 (20.8%) | 288 (10.4%) | 877 (10.6%) | 0.00 |
| **Co-morbidities** | | | | | |
| Time Limited: Minor | | | | | |
| No | 1,816 (64.7%) | 284,858 (77.1%) | 1,807 (65.2%) | 5,482 (66.7%) | 0.02 |
| Yes | 992 (35.3%) | 84,392 (22.9%) | 965 (34.8%) | 2,735 (33.3%) | 0.02 |
| Time Limited: Minor-Primary Infections | | | | | |
| No | 1,628 (58.0%) | 208,304 (56.4%) | 1,620 (58.4%) | 4,931 (60.0%) | 0.03 |
| Yes | 1,180 (42.0%) | 160,946 (43.6%) | 1,152 (41.6%) | 3,286 (40.0%) | 0.03 |
| Time Limited: Major | | | | | |
| No | 2,631 (93.7%) | 348,587 (94.4%) | 2,598 (93.7%) | 7,707 (93.8%) | 0 |
| Yes | 177 (6.3%) | 20,663 (5.6%) | 174 (6.3%) | 510 (6.2%) | 0 |
| Time Limited: Major-Primary Infections | | | | | |
| No | 2,247 (80.0%) | 337,223 (91.3%) | 2,243 (80.9%) | 6,720 (81.8%) | 0.01 |
| Yes | 561 (20.0%) | 32,027 (8.7%) | 529 (19.1%) | 1,497 (18.2%) | 0.01 |
| Allergies | | | | | |
| No | 2,638 (93.9%) | 344,475 (93.3%) | 2,607 (94.0%) | 7,719 (93.9%) | 0.01 |
| Yes | 170 (6.1%) | 24,775 (6.7%) | 165 (6.0%) | 498 (6.1%) | 0.01 |
| Asthma | | | | | |
| No | 2,679 (95.4%) | 349,850 (94.7%) | 2,643 (95.3%) | 7,842 (95.4%) | 0.01 |
| Yes | 129 (4.6%) | 19,400 (5.3%) | 129 (4.7%) | 375 (4.6%) | 0.01 |
| Likely to Recur: Discrete | | | | | |
| No | 1,878 (66.9%) | 261,394 (70.8%) | 1,859 (67.1%) | 5,589 (68.0%) | 0.01 |
| Yes | 930 (33.1%) | 107,856 (29.2%) | 913 (32.9%) | 2,628 (32.0%) | 0.01 |
| Likely to Recur: Discrete-Infections | | | | | |
| No | 2,319 (82.6%) | 296,619 (80.3%) | 2,290 (82.6%) | 6,917 (84.2%) | 0.04 |
| Yes | 489 (17.4%) | 72,631 (19.7%) | 482 (17.4%) | 1,300 (15.8%) | 0.04 |
| Likely to Recur: Progressive | | | | | |
| No | 2,748 (97.9%) | 360,985 (97.8%) | 2,714 (97.9%) | 8,046 (97.9%) | 0 |
| Yes | 60 (2.1%) | 8,265 (2.2%) | 58 (2.1%) | 171 (2.1%) | 0 |
| Chronic Medical: Stable | | | | | |
| No | 1,785 (63.6%) | 248,173 (67.2%) | 1,764 (63.6%) | 4,898 (59.6%) | 0.09 |
| Yes | 1,023 (36.4%) | 121,077 (32.8%) | 1,008 (36.4%) | 3,319 (40.4%) | 0.09 |
| Chronic Medical: Unstable | | | | | |
| No | 2,298 (81.8%) | 310,477 (84.1%) | 2,270 (81.9%) | 6,621 (80.6%) | 0.03 |
| Yes | 510 (18.2%) | 58,773 (15.9%) | 502 (18.1%) | 1,596 (19.4%) | 0.03 |

*(Continued)*

**Table 1.** (*Continued*)

| Variables | Before matching | | After matching | | |
|---|---|---|---|---|---|
| | Exposed individuals (n = 2,808) | Unexposed individuals (n = 369,250) | Exposed individuals (n = 2,772) | Unexposed individuals (n = 8,217) | Weighted Standardized Differences |
| Chronic Specialty: Stable-Orthopedic | | | | | |
| No | 2,712 (96.6%) | 361,174 (97.8%) | 2,676 (96.5%) | 7,921 (96.4%) | 0.01 |
| Yes | 96 (3.4%) | 8,076 (2.2%) | 96 (3.5%) | 296 (3.6%) | 0.01 |
| Chronic Specialty: Stable-Ear, Nose, Throat | | | | | |
| No | 2,735 (97.4%) | 361,681 (98.0%) | 2,702 (97.5%) | 8,000 (97.4%) | 0.01 |
| Yes | 73 (2.6%) | 7,569 (2.0%) | 70 (2.5%) | 217 (2.6%) | 0.01 |
| Chronic Specialty: Stable-Eye | | | | | |
| No | 2,636 (93.9%) | 352,278 (95.4%) | 2,603 (93.9%) | 7,719 (93.9%) | 0 |
| Yes | 172 (6.1%) | 16,972 (4.6%) | 169 (6.1%) | 498 (6.1%) | 0 |
| Chronic Specialty: Unstable-Orthopedic | | | | | |
| No | 2,765 (98.5%) | 363,359 (98.4%) | 2,729 (98.4%) | 8,058 (98.1%) | 0.03 |
| Yes | 43 (1.5%) | 5,891 (1.6%) | 43 (1.6%) | 159 (1.9%) | 0.03 |
| Chronic Specialty: Unstable-Eye | | | | | |
| No | 2,622 (93.4%) | 347,805 (94.2%) | ≥ 99.8%* | ≥ 99.9%* | 0 |
| Yes | 186 (6.6%) | 21,445 (5.8%) | NR | NR | 0 |
| Dermatologic | | | | | |
| No | 2,271 (80.9%) | 319,310 (86.5%) | 2,588 (93.4%) | 7,674 (93.4%) | 0.01 |
| Yes | 537 (19.1%) | 49,940 (13.5%) | 184 (6.6%) | 543 (6.6%) | 0.01 |
| Injuries/Adverse Effects: Minor | | | | | |
| No | 1,980 (70.5%) | 286,652 (77.6%) | 2,248 (81.1%) | 6,714 (81.7%) | 0 |
| Yes | 828 (29.5%) | 82,598 (22.4%) | 524 (18.9%) | 1,503 (18.3%) | 0 |
| Injuries/Adverse Effects: Major | | | | | |
| No | 2,247 (80.0%) | 305,865 (82.8%) | 1,965 (70.9%) | 5,859 (71.3%) | 0 |
| Yes | 561 (20.0%) | 63,385 (17.2%) | 807 (29.1%) | 2,358 (28.7%) | 0 |
| Psychosocial: Time Limited, Minor | | | | | |
| No | 2,700 (96.2%) | 354,306 (96.0%) | 2,219 (80.1%) | 6,606 (80.4%) | 0.02 |
| Yes | 108 (3.8%) | 14,944 (4.0%) | 553 (19.9%) | 1,611 (19.6%) | 0.02 |
| Psychosocial: Recurrent or Persistent, Stable | | | | | |
| No | 2,184 (77.8%) | 293,454 (79.5%) | 2,665 (96.1%) | 7,877 (95.9%) | 0.02 |
| Yes | 624 (22.2%) | 75,796 (20.5%) | 107 (3.9%) | 340 (4.1%) | 0.02 |
| Psychosocial: Recurrent or Persistent, Unstable | | | | | |
| No | 2,714 (96.7%) | 351,916 (95.3%) | 2,156 (77.8%) | 6,341 (77.2%) | 0.02 |
| Yes | 94 (3.3%) | 17,334 (4.7%) | 616 (22.2%) | 1,876 (22.8%) | 0.02 |
| Signs/Symptoms: Minor | | | | | |
| No | 1,770 (63.0%) | 244,578 (66.2%) | 2,679 (96.6%) | 7,969 (97.0%) | 0.02 |
| Yes | 1,038 (37.0%) | 124,672 (33.8%) | 93 (3.4%) | 248 (3.0%) | 0.02 |
| Signs/Symptoms: Uncertain | | | | | |
| No | 1,323 (47.1%) | 201,714 (54.6%) | 1,756 (63.3%) | 5,310 (64.6%) | 0.01 |
| Yes | 1,485 (52.9%) | 167,536 (45.4%) | 1,016 (36.7%) | 2,907 (35.4%) | 0.01 |
| Signs/Symptoms: Major | | | | | |
| No | 1,999 (71.2%) | 274,961 (74.5%) | 1,316 (47.5%) | 3,880 (47.2%) | 0.02 |

(*Continued*)

**Table 1.** (Continued)

| Variables | Before matching | | After matching | | |
|---|---|---|---|---|---|
| | Exposed individuals (n = 2,808) | Unexposed individuals (n = 369,250) | Exposed individuals (n = 2,772) | Unexposed individuals (n = 8,217) | Weighted Standardized Differences |
| Yes | 809 (28.8%) | 94,289 (25.5%) | 1,456 (52.5%) | 4,337 (52.8%) | 0.02 |
| Discretionary | | | | | |
| No | 2,319 (82.6%) | 317,193 (85.9%) | 1,978 (71.4%) | 5,926 (72.1%) | 0.01 |
| Yes | 489 (17.4%) | 52,057 (14.1%) | 794 (28.6%) | 2,291 (27.9%) | 0.01 |
| See and Reassure | | | | | |
| No | 2,709 (96.5%) | 358,851 (97.2%) | 2,292 (82.7%) | 6,773 (82.4%) | 0.02 |
| Yes | 99 (3.5%) | 10,399 (2.8%) | 480 (17.3%) | 1,444 (17.6%) | 0.02 |
| Prevention/Administrative | | | | | |
| No | 1,857 (66.1%) | 247,107 (66.9%) | 2,676 (96.5%) | 7,900 (96.1%) | 0.03 |
| Yes | 951 (33.9%) | 122,143 (33.1%) | 96 (3.5%) | 317 (3.9%) | 0.03 |
| Malignancy | | | | | |
| No | 2,578 (91.8%) | 345,125 (93.5%) | 1,838 (66.3%) | 5,570 (67.8%) | 0.03 |
| Yes | 230 (8.2%) | 24,125 (6.5%) | 934 (33.7%) | 2,647 (32.2%) | 0.03 |
| Pregnancy | | | | | |
| No | 2,748 (97.9%) | 358,134 (97.0%) | 2,546 (91.8%) | 7,480 (91.0%) | 0.05 |
| Yes | 60 (2.1%) | 11,116 (3.0%) | 226 (8.2%) | 737 (9.0%) | 0.05 |
| Dental | | | | | |
| No | 2,770 (98.6%) | 362,489 (98.2%) | 2,712 (97.8%) | 8,095 (98.5%) | 0.02 |
| Yes | 38 (1.4%) | 6,761 (1.8%) | 60 (2.2%) | 122 (1.5%) | 0.02 |

IQR, interquartile range; NR, not report due to small cell; SD, standard deviation

* Cannot be exact due to small cells

Note: Due to small cells, the unmatched individuals were not displayed in this table, but are described in summary in the Results.

## Health outcomes

Within 6-months post-diagnosis, 4.4% of infected individuals sought healthcare services for these conditions similar to known LD sequelae compared to 0.4% in the general population (Table 4). A small proportion of cases (0.7%) required healthcare services for multiple conditions similar to known LD sequelae.

Within 1-year post-diagnosis, exposed individuals were still more likely to seek healthcare services, and 1% of cases developed Lyme meningitis. The RR (95% CI) for arthritis, cardiac abnormalities, cognitive sequelae, dermatologic sequelae, nerve palsies, physical sequelae and polyneuropathy was 4.42 (3.02, 6.47), 2.12 (1.09, 4.11), 1.69 (0.50, 5.79), 5.60 (2.50, 12.55), 62.04 (19.57, 196.64), 2.01 (1.34, 3.02), and 23.72 (2.98, 188.92), respectively. There were 16 cases of Lyme meningitis in infected individuals vs. 0 cases in uninfected individuals. The RR could not be calculated (Table 4).

Infected children and adolescents $\leq$ 18 years of age (n = 378) compared to unexposed (n = 1,119) had increased risks for healthcare visits relating to arthritis (RR 11.70, 95% CI: 5.99–22.87), cardiac abnormalities (RR 3.46, 95% CI: 1.17–10.25), nerve palsies (RR 26.65, 95% CI: 6.22–114.1), physical sequelae (RR 2.22, 95% CI: 1.66–3.08), and Lyme meningitis (11 vs. 0 events). Infected adults (>18 years) (n = 2,394) compared to unexposed (n = 7,098) were at increased risk for healthcare visits relating to arthritis (RR 1.26, 95% CI: 1.05–1.53), nerve palsies (RR 28.81, 95% CI: 13.25–62.65), physical sequelae (RR 1.12, 95% CI: 1.01–1.25), dermatologic conditions (RR 1.97, 95% CI: 1.58–2.47), polyneuropathy (RR 2.36, 95% CI: 1.52–3.65), and Lyme meningitis (16 vs. 0 events).

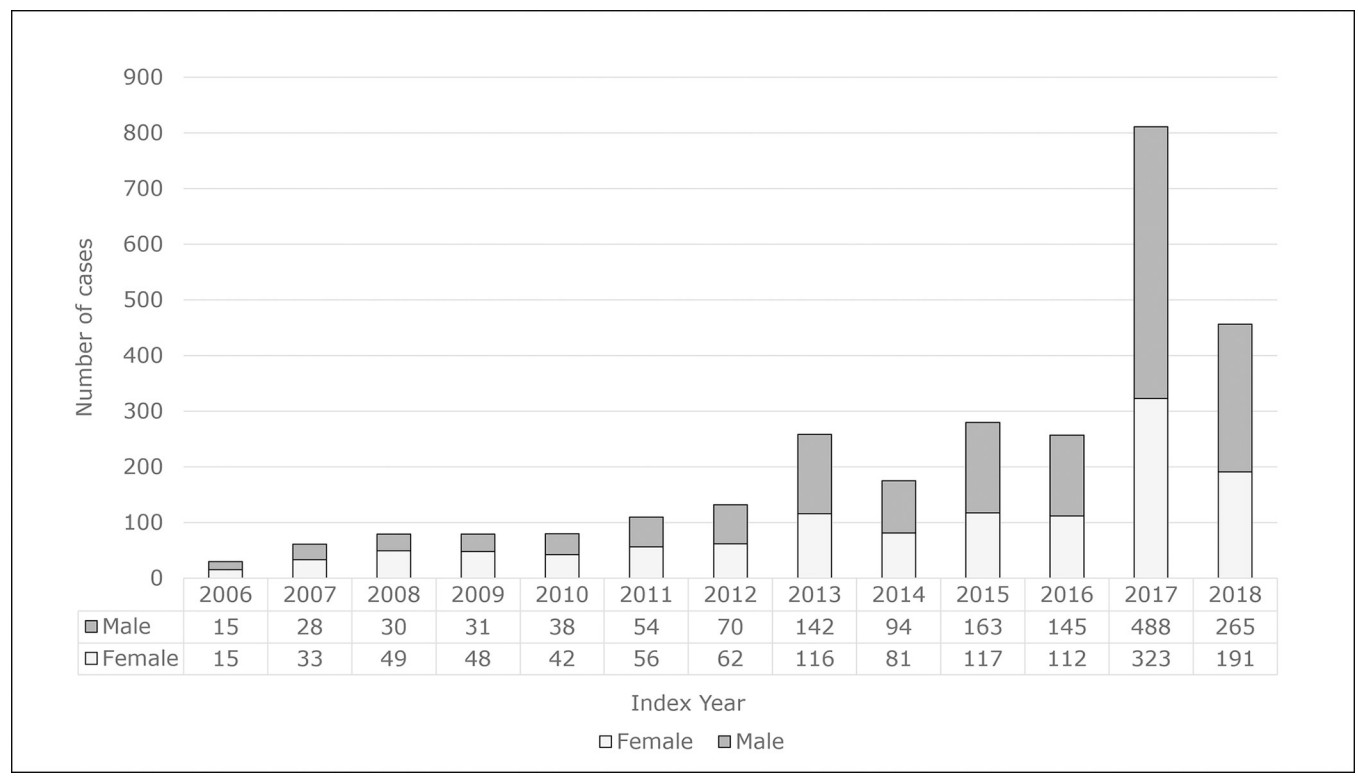

**Fig 2. Laboratory-confirmed LD cases (n = 2,808) by index year and sex.**

## Discussion

Laboratory-confirmed LD resulted in increased healthcare costs 10-days prior to diagnosis, and up to 6 months post-infection. Mean attributable costs were highest 30-days post-infection and in the 10 days prior to diagnosis. In the acute care phase (30 days post-diagnosis), approximately 2% of cases were hospitalized with a mean length of stay of 5 days, but 58% of attributable costs were related to hospitalization, suggesting that while uncommon, hospitalizations contribute substantially to LD economic burden. In a recent Lyme borreliosis costing study from Poland [23], hospitalizations contributed to 67% of total LD costs, similar to the 58% from our study. Physician and ED visit costs were the next highest costs in the acute care

**Table 2. Healthcare resource use for exposed cohort (n = 2,808).**

| Time period | Frequency (percentage of total cases N = 2,808) | | |
|---|---|---|---|
| | New ED visits* | Hospitalizations* with LD (A69.2) | Hospitalizations* with LD (A69.2) as MRDX |
| Pre-diagnosis | 30 (1.1%) | < 6 (< 0.2%) | < 6 (< 0.2%) |
| 1-10d post-diagnosis | 181 (6.4%) | 43 (1.6%) | 27 (1.0%) |
| 11-30d post-diagnosis | 223 (7.9%) | 52 (1.8%) | 39 (1.4%) |
| 31-90d post-diagnosis | 115 (4.1%) | 33 (1.2%) | 19 (0.7%) |
| >90d post-diagnosis | 28 (1.0%) | 10 (0.3%) | < 6 (< 0.2%) |
| Mean (SD) LOS | 3.21 (2.91) hours | 7.32 (9.11) days | 5.52 (4.19) days |

*Number of cases requiring ED visits or hospitalization, regardless of number of visits.

ED, emergency department; LD, Lyme disease; LOS, length of stay; MRDX, most responsible diagnosis; SD, standard deviation

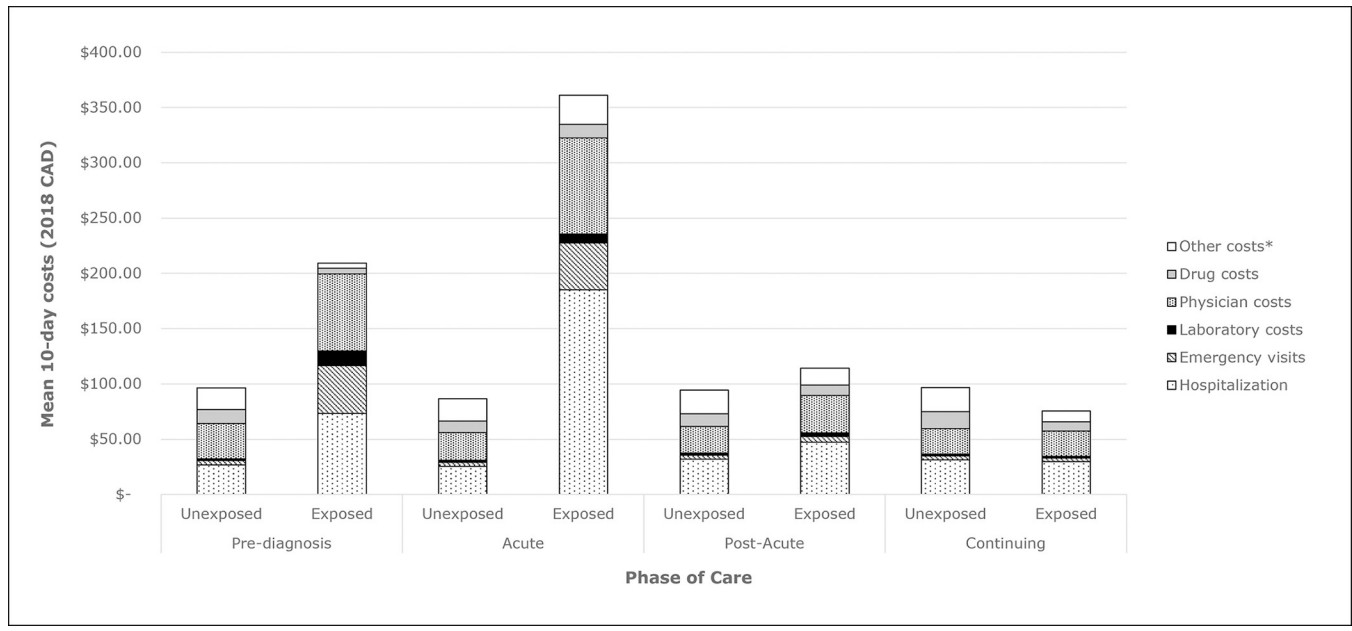

**Fig 3. Mean 10-day standardized costs for Lyme disease phase-of-care between pre-diagnosis to continuing care.**

phase. Drug costs were low from a healthcare payer perspective because the dataset only captures drug costs for individuals in outpatient settings who are ODB eligible (i.e., ≥ 65 years of age) [24].

In the post-acute care phase (2 to 6 months post-diagnosis), healthcare costs attributable to LD were potentially driven by those who develop conditions similar to known LD sequelae, albeit a fraction of the costs incurred during the acute care phase. Within 6-months post-diagnosis, there was increased risk for individuals infected with LD to seek healthcare services for conditions similar to known LD sequelae: arthritis, cardiac sequelae, nerve palsies, dermatologic sequelae, physical sequelae (headaches, ataxia, muscle pain, arthralgia), Lyme meningitis, and polyneuropathy. Due to unavailable billing codes specific to LD, our results can only be

**Table 3. Standardized 10-day mean attributable costs by LD stage[a].**

| Type | Mean 10-day attributable costs (2018 CAD) | | | | |
| --- | --- | --- | --- | --- | --- |
| | **Pre-diagnosis** | **Acute** | **Post-acute** | **Continuing** | **Predeath** |
| **Hospitalization costs [b]** | $46 ($26, $67) | $160 ($127, $192) | $15 ($4, $26) | -$2 (-$9, $6) | $479 (-$1,163, $2,122) |
| **Emergency visit costs** | $39 ($33, $45) | $39 ($35, $42) | $2 ($1, $2) | $0 (-$1, $0) | -$8 (-$36, $21) |
| **Laboratory costs** | $12 ($10, $13) | $6 ($6, $7) | $1 ($1, $1) | $0 ($0, $0) | -$1 (-$4, $2) |
| **Physician costs [b]** | $38 ($30, $45) | $62 ($55, $69) | $10 ($7, $13) | $0 (-$2, $1) | -$55 (-$186, $76) |
| **Drug costs** | -$7 (-$13, -$2) | $2 (-$1, $5) | -$2 (-$5, $1) | -$7 (-$9, -$4) | $61 (-$28, $150) |
| **Other costs [b]** | -$15 (-$21, -$9) | $6 (-$3, $16) | -$6 (-$13, $1) | -$12 (-$17, -$8) | -$87 (-$411, $237) |
| **Total costs** | $113 ($81, $144) | $275 ($231, $319) | $20 ($2, $37) | -$22 (-$33, -$10) | $391 (-$1,152, $1,934) |

[a] Pre-diagnosis phase: 10 days prior to index date; acute phase: index to 30 day post-index date; post-acute phase: 31 to 180 days (6 months) post index date; continuing care phase: Remaining observation time between 6 months post-index and end of follow-up or predeath phase if individual died; predeath phase: 90 days prior to all-cause death.

[b] Hospitalization costs include only salaried physician services; physician services provided to inpatients are included in physician costs.

[c] Other costs include: long-term care, continuing care, rehabilitation, mental health, dialysis, cancer, and home care

**Table 4. Conditions similar to known LD sequelae at different follow-up post-index dates stratified by age and sex.**

| Type of condition | 6 months follow-up post-index | | | 1-year follow-up post-index | | | 3-years follow-up post-index | | |
|---|---|---|---|---|---|---|---|---|---|
| | Unexposed (n = 8,217) | Exposed (n = 2,772) | RR (95% CI) | Unexposed (n = 8,217) | Exposed (n = 2,772) | RR (95% CI) | Unexposed (n = 8,217) | Exposed (n = 2,772) | RR (95% CI) |
| Arthritis | 19 (0.2%) | 54 (1.9%) | 8.39 (5.03, 13.9)* | 42 (0.5%) | 63 (2.3%) | 4.42 (3.02, 6.47)* | 336 (4.1%) | 178 (6.4%) | 1.57 (1.31, 1.87)* |
| Cardiac | < 6 (< 0.1%) | 9 (0.3%) | 6.67 (2.06, 21.6)* | 21 (0.3%) | 15 (0.5%) | 2.12 (1.09, 4.11)* | 338 (4.1%) | 128 (4.6%) | 1.12 (0.92, 1.36) |
| Cognitive | NA | NA | NA | 7 (0.1%) | < 6 (< 0.1%) | 1.69 (0.50, 5.79) | 218 (2.7%) | 91 (3.3%) | 1.24 (0.98, 1.57) |
| Dermatologic | < 6 (< 0.1%) | 11 (0.4%) | 10.88 (3.04, 38.9)* | 9 (0.1%) | 17 (0.6%) | 5.60 (2.50, 12.55)* | 185 (2.3%) | 123 (4.4%) | 1.97 (1.59, 2.45)* |
| Meningitis | 0 (0%) | 13 (0.5%) | NR* | 0 (0%) | 16 (0.6%) | NR* | 0 (0%) | 27 (1.0%) | NR* |
| Nerve palsies | < 6 (< 0.1%) | 58 (2.1%) | 171.84 (23.96 1,232.5)* | < 6 (< 0.1%) | 58 (2.1%) | 62.04 (19.57, 196.64)* | 9 (0.1%) | 86 (3.1%) | 28.32 (14.27, 56.20)* |
| Physical | 9 (0.1%) | 14 (0.5%) | 4.61 (2.00, 10.6)* | 56 (0.7%) | 38 (1.4%) | 2.01 (1.34, 3.02)* | 1,015 (12.4%) | 411 (14.8%) | 1.20 (1.08, 1.32)* |
| Polyneuropathy | 0 (0.0%) | 6 (0.2%) | NR* | < 6 (< 0.1%) | 8 (0.3%) | 23.72 (2.98, 188.92)* | 45 (0.5%) | 36 (1.3%) | 2.36 (1.52, 3.65)* |
| **Sequelae count** | | | | | | | | | |
| Single (n = 1) | 32 (0.4%) | 122 (4.4%) | - | 112 (1.4%) | 159 (5.7%) | - | 1,196 (14.6%) | 533 (19.2%) | - |
| Multiple (n ≥ 2) | < 6 (< 0.1%) | 20 (0.7%) | - | 17 (0.1%) | 30 (1.1%) | - | 433 (5.2%) | 245 (8.7%) | - |

*p-value < 0.05

CI, confidence interval; LD, Lyme disease; NA, not available to be calculated due to zero counts; RR, relative risk

interpreted to suggest an increased attributable risk for healthcare visits related to LD sequelae as reported in the literature, but are not distinct reports of LD sequelae (e.g., Lyme arthritis, Lyme carditis). We excluded individuals who had healthcare visits for similar LD-related sequalae 1-year prior to their LD infection in order to increase the likelihood of these health-care visits being attributable to the LD-infection.

In the continuing care phase, there were no costs attributable to LD, suggesting that LD may not generally result in long-term use of healthcare resources. Even though we identified a balanced matched cohort, there may be remaining unmeasurable confounders or high-cost users in the control group. Another hypothesis is that individuals who were infected with LD still have unobserved social or physical confounders that contribute to their risk of LD infection (e.g., increased leisure time, affinity to outdoors) and an inherently healthier profile given risk of LD infection is related to time spent outdoors where vectors carrying *B. burgdorferi* are present to transmit the bacteria [25]. Lastly, patients infected with LD may be seeking alternative care outside of the healthcare system, which would not have been captured in this analysis [26, 27].

In comparison to the earlier study from our colleagues using LD cases from 2006 to 2013 [6], the mean age and proportion of male cases were lower, hospitalizations within 30-days post-index was slightly lower (3.4% vs. 4.7%), and attributable 10-day costs for LD in the 30-days post index were similar ($275 vs. $277). Our study estimated increased healthcare resource use prior to diagnosis with an attributable cost of $113 in those 10 days due to diagnostic workup.

Our study is subject to several limitations. Our analysis used a cohort of individuals with LD from a linked laboratory and reportable disease dataset. We were unable to report costs based on LD stage (i.e., early localized vs. disseminated LD) due to limited data collected on

LD stage at diagnosis, symptoms experienced, or time from initial symptom onset. These datasets are vulnerable to underreporting. The use of a PHO test results may inadvertently introduce selection and sampling bias as out-of-province testing, false negatives, or lack of diagnostic confirmation were excluded from the analysis. Those who are typically able to receive test results are more likely to receive timely and adequate healthcare and thus, the exclusion of those without positive test results may underestimate the overall economic burden of the disease. However, by using this LD case definition, we ensured consistent diagnosis across all cases to provide best available evidence to accurately estimate LD burden.

The lack of LD billing code limits our ability to identify those with PTLDS, and may underestimate the risk of LD sequelae. Despite these limitations, we used billing codes reviewed by clinical microbiologists and infectious disease specialists to report relative risks for conditions similar to known LD sequelae in a Canadian context. Lastly, the attributable economic burden of LD may be underreported due to dataset limitations. While the non-medical costs of LD have been recently studied in Belgium and the United States [28, 29], this analysis on economic burden only focuses on the healthcare costs and does not capture societal costs that may arise due to long-term sequelae and PTLDS such as caregiver costs, wellbeing costs, out-of-pocket costs, and productivity loss.

Despite these limitations, our study provides comprehensive estimates of the economic and health burden of laboratory-confirmed LD using a cohort of over 2,500 cases over 13 years in a province with the second highest Canadian incidence rates. We used propensity score matching to reduce potential observable confounders. These cost estimates can further the understanding on LD burden, use of healthcare resources throughout different phases of the disease, and facilitate cost-effectiveness analysis on interventions (vaccination programs) in high-risk areas.

## Conclusion

Direct healthcare costs attributable to lab-confirmed LD are highest in the 30 days post-diagnosis, and 10 days prior to diagnosis. Post-diagnosis, LD increased individuals' use of healthcare services for conditions similar to known LD sequelae. This study highlighted the importance of being able to identify LD infections, and can support future decision-making around public health interventions.

## Supporting information

**S1 File. Case definitions for Lyme disease in Ontario.**
(DOCX)

**S1 Table. Variables included in propensity score regression and matching.**
(DOCX)

**S2 Table. Cost variable definitions and source of data.**
(DOCX)

**S3 Table. ICD-10 and OHIP codes to identify Lyme disease and potentially related sequelae.**
(DOCX)

**S4 Table. RECORD checklist.**
(DOCX)

**S5 Table. Laboratory-confirmed LD cases by age and sex (n = 2,808).**
(DOCX)

**S6 Table. Baseline characteristics of matched cohort at predeath.**
(DOCX)

**S7 Table. Mean phase-of-care costs for matched exposed and unexposed individuals.**
(DOCX)

## Acknowledgments

Parts of this material are based on data and information compiled and provided by MOH, and CIHI. The analyses, conclusions, opinions and statements expressed herein are solely those of the authors and do not reflect those of the funding or data sources; no endorsement is intended or should be inferred. Parts of this material are based on data and/or information compiled and provided by CIHI. However, the analyses, conclusions, opinions and statements expressed in the material are those of the author(s), and not necessarily those of CIHI. The analyses, conclusions, views, opinions and statements expressed in this article are those of the author(s) and do not necessarily represent those of, or reflect, the official position of Public Health Ontario.

We would like to acknowledge Andrew Mendlowitz, Li Bai, James Jung, Alexander Kopp, and Alex Marchand-Austin for guidance on the analyses.

## Author Contributions

**Conceptualization:** Stephen Mac, Beate Sander.

**Data curation:** Stephen Mac, Beate Sander.

**Formal analysis:** Stephen Mac, Gerald Evans, Eleanor Pullenayegum, Samir N. Patel, Beate Sander.

**Funding acquisition:** Stephen Mac, Beate Sander.

**Methodology:** Stephen Mac, Gerald Evans, Eleanor Pullenayegum, Samir N. Patel, Beate Sander.

**Supervision:** Beate Sander.

**Writing – original draft:** Stephen Mac, Beate Sander.

**Writing – review & editing:** Stephen Mac, Gerald Evans, Eleanor Pullenayegum, Samir N. Patel, Beate Sander.

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
