## [Decision Letter · Decision Letter 0]

6 Apr 2023

PONE-D-23-05674Healthcare Costs and Outcomes Associated with Laboratory-confirmed Lyme disease in Ontario, Canada: A Population-based Cohort Study

PLOS ONE

Dear Dr. Stephen Mac,

Thank you for submitting your manuscript to PLOS ONE. After careful consideration, the reviewers feel that certain specific issues must be addressed before the manuscript can be accepted. Please see the comments below. We invite you to submit a revised version of the manuscript that addresses the points raised during the review process.

We look forward to receiving your revised manuscript.

Kind regards,

Shuo-Yan Gau

Academic Editor

PLOS ONE

Journal Requirements:

Reviewers' comments:

Reviewer's Responses to Questions

**Comments to the Author**

1. Is the manuscript technically sound, and do the data support the conclusions?

Reviewer #1: Yes

Reviewer #2: Yes

2. Has the statistical analysis been performed appropriately and rigorously? 

Reviewer #1: Yes

Reviewer #2: Yes

3. Have the authors made all data underlying the findings in their manuscript fully available?

Reviewer #1: Yes

Reviewer #2: Yes

4. Is the manuscript presented in an intelligible fashion and written in standard English?

Reviewer #1: Yes

Reviewer #2: Yes

5. Review Comments to the Author

Reviewer #1: I appreciate that this study has provided the associations between the health burden of LD and its utilization of healthcare resources in Ontario. This manuscript is well organized, and the topic has good merit to discuss. However, I am curious why the study period is from 2006 to 2018. Additionally, Figure 3 appears a bit cluttered and visually confusing.

Reviewer #2: The manuscript provided rigorous and extensive analyses on the direct medical costs of Lyme disease. However, I would like to suggest the authors to provide some clarifications

- Page 6 line 108-109: Please provide more details about how the unexposed participants were idenified and deemed as uninfected (ie. laboratory-confirmed uninfected participants).

- The costs should be accounted for the staging of Lyme disease, namely erythema migrans and disseminated lyme disease.

- I strongly recommend the authors include non-medical costs of Lyme disease, such as absenteeism and presenteeism. Examples of recent studies including non-medical costs of Lyme disease are as follows; BMC Public Health. 2022 Nov 28;22(1):2194. doi: 10.1186/s12889-022-14380-6. and Emerg Infect Dis. 2022 Jun;28(6):1170-1179. doi: 10.3201/eid2806.211335. However, I completely leave the decision whether to perform non-medical cost analyses to the authors.

- There were several "Error! Reference source not found" in the manuscript. Please correct them.

6. PLOS authors have the option to publish the peer review history of their article (what does this mean?). If published, this will include your full peer review and any attached files.

Reviewer #1: No

Reviewer #2: No

---

## [Author Response · Author response to Decision Letter 0]

21 Apr 2023

Stephen Mac

Institute of Health Policy, Management, and Evaluation

University of Toronto

155 College St., 4th floor

Toronto, ON M5T 3M6

Email: sm.mac@mail.utoronto.ca

To: Dr. Shuo-Yan Gau

Academic Editor

PLOS ONE

April 21, 2023

RE: PONE-D-23-05674 “Healthcare Costs and Outcomes Associated with Laboratory-confirmed Lyme disease in Ontario, Canada: A Population-based Cohort Study”

Dear Dr. Gau,

Thank you for taking the time to review and move forward with our manuscript. We have addressed all comments and suggestions in this resubmission to PLOS ONE. Please find enclosed our revised manuscript. We have addressed the additional requirements per your request and ensured that our manuscript meets PLOS ONE’s style requirements. 

Regarding data availability, we outline this in the Footnotes under Data Availability Statement that is standard for all studies conducted at ICES: “The dataset from this study is held securely in coded form at ICES. While legal data sharing agreements between ICES and data providers (e.g., healthcare organizations and government) prohibit ICES from making the dataset publicly available, access may be granted to those who meet pre-specified criteria for confidential access, available at www.ices.on.ca/DAS (email: das@ices.on.ca). The full dataset creation plan and underlying analytic code are available from the authors upon request, understanding that the computer programs may rely upon coding templates or macros that are unique to ICES and are therefore either inaccessible or may require modification.”

Following this letter, we included our detailed response (with page numbers corresponding to the tracked manuscript) to all comments for your review. 

Thank you for your time and consideration. We look forward to the final decision.

Sincerely,

Stephen Mac 

On behalf of all authors below

Stephen Mac PhD

Institute of Health Policy, Management and Evaluation, University of Toronto, Toronto, Canada

Toronto Health Economics and Technology Assessment (THETA) Collaborative, University Health Network, Toronto, Canada

Gerald A. Evans MD FRCPC 

Department of Medicine, Queen’s University, Kingston, Canada

Samir N. Patel PhD FCCM

Laboratory Medicine and Pathobiology, University of Toronto, Toronto, Canada

Public Health Ontario, Toronto, Canada

Eleanor M. Pullenayegum PhD

Dalla Lana School of Public Health, University of Toronto, Toronto, Canada

The Hospital for Sick Children, Toronto, Canada

Beate Sander PhD

Institute of Health Policy, Management and Evaluation, University of Toronto, Toronto, Canada

Toronto Health Economics and Technology Assessment (THETA) Collaborative, University Health Network, Toronto, Canada

ICES, Toronto, Canada

Public Health Ontario, Toronto, Canada

 

REVIEWER #1 COMMENTS

Comment: I appreciate that this study has provided the associations between the health burden of LD

and its utilization of healthcare resources in Ontario. This manuscript is well organized, and the topic has

good merit to discuss. However, I am curious why the study period is from 2006 to 2018. 

Response: Thank you for this comment. The study period is limited from 2006 to 2018 as this was the duration of data that was available at the time this study was initiated in 2020. Since the laboratory data comes from Public Health Ontario, there are various timely processes required to de-identify, clean, and transfer the data to ICES in Ontario for linkage to administrative data and for subsequent analysis. Unfortunately, this data cut cannot be updated for this study at this point due to lack of resources. However, we are confident that this cohort, which includes 13 years of patients, provides an appropriate sample size to provide an estimate of the health outcomes and economic burden associated with lab-confirmed Lyme disease in Ontario, Canada.

Comment: Additionally, Figure 3 appears a bit cluttered and visually confusing.

Response: We agree that this figure may feel clustered and so we have focused on the four phases: pre-diagnosis, acute, post-acute and continuing care. We have also included a description of Figure 3 in the manuscript and how it can be interpreted on page 15 (lines 220-222). The Figure 3 caption has been updated to “Figure 3. Mean 10-day standardized costs for Lyme disease phase-of-care between pre-diagnosis to continuing care” on page 15 (lines 226-227).

REVIEWER #2 COMMENTS

Comment: Page 6 line 108-109: Please provide more details about how the unexposed participants were identified

and deemed as uninfected (i.e., laboratory-confirmed uninfected participants).

Response: We described in detail how unexposed individuals were identified in the manuscript on page 5 (lines 93-95): “Individuals neither infected with LD nor had a PHO serologic test record were selected from the Registered Persons Database (RPDB) for matching. We excluded individuals with negative PHO test results to avoid potential confounding healthcare seeking behaviours.” This description in the original manuscript seems out of place and we have taken your suggestion and moved the details to page 6 (lines 118-120) to describe the unexposed individual selection before describing the matching technique. 

Comment: The costs should be accounted for the staging of Lyme disease, namely erythema migrans and

disseminated Lyme disease. 

Response: This is a very good comment. We would have also preferred to present the analysis based on the staging of Lyme disease (i.e., early localized, early disseminated, late disseminated). However, the variables collected within the laboratory (Public Health Ontario) and reportable disease dataset (iPHIS) do not identify the LD stage, symptoms experienced, or time from symptom onset. Therefore, we elected to present these findings as costs within standardized phases post-diagnosis. We have acknowledged this limitation in the Discussion section on page 20 (lines 308-310).

Comment: I strongly recommend the authors include non-medical costs of Lyme disease, such as absenteeism and

presenteeism. Examples of recent studies including non-medical costs of Lyme disease are as follows; BMC Public Health. 2022 Nov 28;22(1):2194. doi: 10.1186/s12889-022-14380-6. and Emerg Infect Dis. 2022 Jun;28(6):1170-1179. doi: 10.3201/eid2806.211335. However, I completely leave the decision whether to perform non-medical cost analyses to the authors.

Response: Thanks for the comment. The societal costs of Lyme disease are important to understand and indeed missing due to the objective of this study, which was to understand the real-world economic burden of Lyme disease to the healthcare system using administrative data. Unfortunately, societal costs are unable to be estimated with health administrative data that we have access to in Canada, as the data collected is solely from the Minister of Health’s perspective and does not capture time off, indirect costs, caregiver burden, transportation used, etc. These costs would need to be estimated via other data sources or more appropriate study designs such as a cross-sectional survey or prospective cohort study, along with applying a human capital or friction cost method. We have highlighted this in our Discussion and further expanded on the recent studies you shared that were completed in Belgium and the United States on page 21 (lines 322-323).

Comment: There were several "Error! Reference source not found" in the manuscript. Please correct them.

Response: Apologies for these formatting errors in the manuscript. We have identified all the cross-reference errors and have corrected them all.

---

## [Decision Letter · Decision Letter 1]

18 May 2023

Healthcare Costs and Outcomes Associated with Laboratory-confirmed Lyme disease in Ontario, Canada: A Population-based Cohort Study

PONE-D-23-05674R1

Dear Dr. Mac,

We’re pleased to inform you that your manuscript has been judged scientifically suitable for publication and will be formally accepted for publication once it meets all outstanding technical requirements.

Kind regards,

Shuo-Yan Gau

Academic Editor

PLOS ONE

Additional Editor Comments (optional):

Reviewers' comments:

Reviewer's Responses to Questions

**Comments to the Author**

1. If the authors have adequately addressed your comments raised in a previous round of review and you feel that this manuscript is now acceptable for publication, you may indicate that here to bypass the “Comments to the Author” section, enter your conflict of interest statement in the “Confidential to Editor” section, and submit your "Accept" recommendation.

Reviewer #1: All comments have been addressed

Reviewer #2: All comments have been addressed

2. Is the manuscript technically sound, and do the data support the conclusions?

Reviewer #1: Yes

Reviewer #2: Yes

3. Has the statistical analysis been performed appropriately and rigorously? 

Reviewer #1: Yes

Reviewer #2: Yes

4. Have the authors made all data underlying the findings in their manuscript fully available?

Reviewer #1: Yes

Reviewer #2: Yes

5. Is the manuscript presented in an intelligible fashion and written in standard English?

Reviewer #1: Yes

Reviewer #2: Yes

6. Review Comments to the Author

Reviewer #1: (No Response)

Reviewer #2: (No Response)

7. PLOS authors have the option to publish the peer review history of their article (what does this mean?). If published, this will include your full peer review and any attached files.

Reviewer #1: No

Reviewer #2: No

---

## [Editor Report · Acceptance letter]

13 Jun 2023

PONE-D-23-05674R1 

Healthcare Costs and Outcomes Associated with Laboratory-confirmed Lyme disease in Ontario, Canada: A Population-based Cohort Study 

Dear Dr. Mac:

I'm pleased to inform you that your manuscript has been deemed suitable for publication in PLOS ONE. Congratulations! Your manuscript is now with our production department. 

Kind regards, 

on behalf of

Mr. Shuo-Yan Gau 

Academic Editor

PLOS ONE